# Reputation effects drive the joint evolution of cooperation and social rewarding

Saptarshi Pal [1] & Christian Hilbe [1]

People routinely cooperate with each other, even when cooperation is costly. To further encourage such pro-social behaviors, recipients often respond by providing additional incentives, for example by offering rewards. Although such incentives facilitate cooperation, the question remains how these incentivizing behaviors themselves evolve, and whether they would always be used responsibly. Herein, we consider a simple model to systematically study the co-evolution of cooperation and different rewarding policies. In our model, both social and antisocial behaviors can be rewarded, but individuals gain a reputation for how they reward others. By characterizing the game's equilibria and by simulating evolutionary learning processes, we find that reputation effects systematically favor cooperation and social rewarding. While our baseline model applies to pairwise interactions in well-mixed populations, we obtain similar conclusions under assortment, or when individuals interact in larger groups. According to our model, rewards are most effective when they sway others to cooperate. This view is consistent with empirical observations suggesting that people reward others to ultimately benefit themselves.

When interacting in groups, individuals regularly encounter social dilemmas. In social dilemmas, individuals may take actions that are detrimental to themselves but beneficial for other group members[1–3]. Such behaviors are usually referred to as cooperation[4–6]. Instances of cooperation are abound[7]. They arise when people do favors[8,9], when animals share food or other commodities[10–12], or when microorganisms produce diffusible public goods[13]. While some of these instances are readily explained by kin selection, especially humans have no difficulty to establish cooperation beyond the narrow scope of their own families[14,15]. To achieve cooperation, humans often modify the exact make-up of their social interactions[16]. For example, to incentivize pro-social behaviors, they may change the strategic nature of interaction by rewarding cooperative behaviors[17–30]. Conversely, to disincentivize defection, they may exert punishment in response to any misbehavior[31–36]. Past work has shown that both, rewards and punishment, can greatly promote cooperative behavior[37,38].

Yet from a theoretical perspective, the introduction of rewards and punishments seems to only shift the problem from explaining why people cooperate to explaining why they reward or punish others. This is the so-called second-order free rider problem[39,40]: everyone prefers cooperation to be incentivized, but people prefer others to bear the respective costs[41,42]. In addition to the second-order free rider problem, both rewards and punishment come with a range of additional problems. For example, there is substantial literature suggesting that punishment can be misused[43–47]. Instead of targeting defectors, subjects in behavioral experiments often engage in counter-punishment[43] or anti-social punishment[44]. As a result, overall payoffs in experiments with punishment often tend to be lower than in experiments without it[48,49] (for an exception, see ref. 33). Most early studies on the evolution of punishment neglect these detrimental forms of punishment. Once these detrimental forms are included, the very same models often predict that cooperation breaks down[45–47]. Stable cooperation either seems to require that certain behaviors cannot be punished[50], or that anti-social punishers bear the risk of gaining a negative reputation[51,52].

While there is by now a rich theoretical literature on punishment[37,38], the evolution of rewarding is somewhat less studied.

---

[1]Max Planck Research Group Dynamics of Social Behavior, Max Planck Institute for Evolutionary Biology, 24306 Plön, Germany.
e-mail: pal@evolbio.mpg.de

This may come as a surprise, as rewards are less susceptible to some of the major drawbacks of sanctions. Rewards cannot be misused for retaliation or spite, nor do they bear the risk of reducing overall welfare[19]. Existing models predict that rewards can promote cooperation and that they are particularly effective in populations with only a few cooperators (such that rewarding those cooperators is relatively cheap[21–24]). These conclusions, however, are (again) based on biased strategy sets[53]. The models assume that while cooperators can be rewarded, defectors cannot. Once defectors can be rewarded, some more recent modeling work on institutionalized rewards suggests that antisocial rewarding may prevail[25–27]. For some parameter combinations, this work shows that defectors do not necessarily learn to use rewards to incentivize cooperation. Rather they learn to reward other defectors. These results point out a more general lack in our understanding of the functional role of rewards. When individuals themselves have the freedom to choose who to reward, which kinds of behavior would they incentivize? In which environments would rewards promote the evolution of cooperation?

In the following, we present a simple game-theoretic model to address these questions on the co-evolution of cooperation and rewarding. In our model, individuals can reward any type of behavior. They can explicitly reward cooperation (social rewarding), reward defection (antisocial rewarding), never reward, or always reward. Importantly, however, the way how individuals reward others can become publicly known with some probability. In particular, similar to earlier models[21–24], people may learn that a given group member tends to reward socially (or anti-socially). Such knowledge allows individuals to react opportunistically. They may cooperate with people who are known to reward cooperators while defecting against those people who either do not reward or who reward defectors. We show that these reputation effects are crucial for the behaviors that evolve. When people's rewarding strategies remain unknown, cooperation and (social) rewarding only emerge in populations with assortment (in which case also defection and antisocial rewarding may emerge). But once individuals can gain a reputation for how they reward others, they systematically prefer to reward socially, and as a result, to cooperate. We first present these results for simple interactions between two individuals. However, as we show further below, similar insights can be obtained when individuals interact in larger groups.

Our model suggests that rewards are most effective when they are used as a strategic means to persuade others to cooperate. It also suggests an interesting asymmetry in how people use rewards. If there are reputational consequences, people have strong incentives to reward pro-social behaviors only. Anti-social rewarding may still evolve, but it requires rather restrictive conditions, such as strong assortment, or that rewards are mutually beneficial for both the rewarded and the rewarder. Overall, we show that in the presence of reputation effects, rewards systematically favor cooperation.

## Results

### A model of cooperation and rewarding in pairwise interactions

For the baseline version of our model, we consider interactions between two individuals, a donor and a recipient (the 'players'). Interactions take place in two stages, as visualized in Fig. 1a. The first stage is the donation stage. Here, the donor decides whether or not to provide a benefit $b > 0$ to the recipient at a cost $c > 0$. We refer to these two possible actions as cooperation and defection, respectively. In the second stage, the recipient decides whether or not to reward the donor. Rewards have a positive effect of $\beta > 0$ on the payoff of the donor, but they reduce the recipient's payoff by $\gamma > 0$. Depending on the actions of the donor, there are four possible rewarding strategies. The recipient can either never reward the donor (NR), reward donors who cooperate (social rewarding, SR), reward donors who defect (anti-social rewarding, AR), or reward unconditionally (UR). The last two options are absent in earlier two-player models of cooperation and rewarding[21,22]. This earlier work asks whether social rewards can promote the evolution of cooperation. In contrast, we ask in which environments the individuals learn to use social forms of rewarding in the first place. To allow for interesting dynamics, we assume that rewards are sufficient to offset the costs of cooperation ($\beta > c$) and that the benefits of cooperation offset the costs of rewarding ($b > \gamma$). If either of these two conditions is violated, we show in Supplementary Note 1 that neither cooperation nor rewarding can emerge.

To define the possible donation strategies in the first stage, we assume that donors learn with some probability $\lambda$ which rewarding strategy the recipient applies. We refer to $\lambda$ as the population's information transmissibility. When $\lambda = 0$, donors lack any information. They make their decision whether to cooperate in ignorance of the recipient's strategy. In contrast, when $\lambda > 0$, there is a chance that donors correctly anticipate how the recipient would react. In that case, donors may act opportunistically. Opportunistic donors cooperate against those recipients who are known to engage in social rewarding, and they defect against all others. These considerations give rise to four strategies in the first stage. Donors may always cooperate ($C$); they may cooperate if the recipient's strategy is unknown and act opportunistically otherwise (opportunistic cooperator, OC); they may defect if the recipient's strategy is unknown and act opportunistically otherwise (opportunistic defector, OD); or they may always defect ($D$). Figure 1b provides an overview of the four strategies of the donor and the recipient. Moreover, in Fig. 1c, we discuss a specific example. There, we consider an interaction between an opportunistic cooperator (OC) who interacts with an anti-social rewarder (AR). We derive the expected payoffs that these two players obtain. Similarly, we can also compute the payoffs for all other fifteen strategy combinations; the respective payoff matrices are displayed as Eqs. (2) and (3) in the "Methods" section.

### Equilibrium analysis

To explore the viability of cooperation and different rewarding strategies, we first characterize all Nash equilibria of the game. Nash equilibria correspond to stable states in which neither the donor nor the recipient can further improve their payoff. We characterize both, all pure Nash equilibria (in which each player chooses a single strategy) and all mixed Nash equilibria (in which players randomize between several strategies). The outcome crucially depends on the recipient's ability to build up a reputation (i.e., on the information transmissibility $\lambda$). When information transmissibility is low, $\lambda < \gamma/b$, there is only one pure Nash equilibrium, ($D$, NR). In this equilibrium, donors defect unconditionally and recipients do not reward. In addition, there is a mixed equilibrium in which donors randomize between the two opportunistic strategies OC and OD, whereas recipients randomize between non-rewarding and social-rewarding, NR and SR. In contrast, once donors are sufficiently likely to learn the recipient's rewarding strategy, $\lambda > \gamma/b$, another pure Nash equilibrium appears, (OC, SR). In this equilibrium, recipients reward socially, and donors cooperate opportunistically. All other equilibria give rise to the same behaviors as the ones we have described above (see Fig. 2a for an overview, and Supplementary Note 1 for all details).

To interpret these results, we note that in the absence of reputation effects ($\lambda = 0$), the interaction between the donor and the recipient takes the form of a simple sequential game with two stages. This game can be solved by backward induction[54]: in the last stage of the game, recipients have no incentive to reward anyone (not even cooperators); as a result, donors have no incentive to cooperate in the first stage. By introducing reputation effects ($\lambda > 0$), the game is transformed. Now there is a chance that donors know the recipient's reaction before having to decide whether to cooperate. Here, it can be beneficial for recipients to gain the reputation of being a social rewarders, and for donors to adapt to the recipient's reputation.

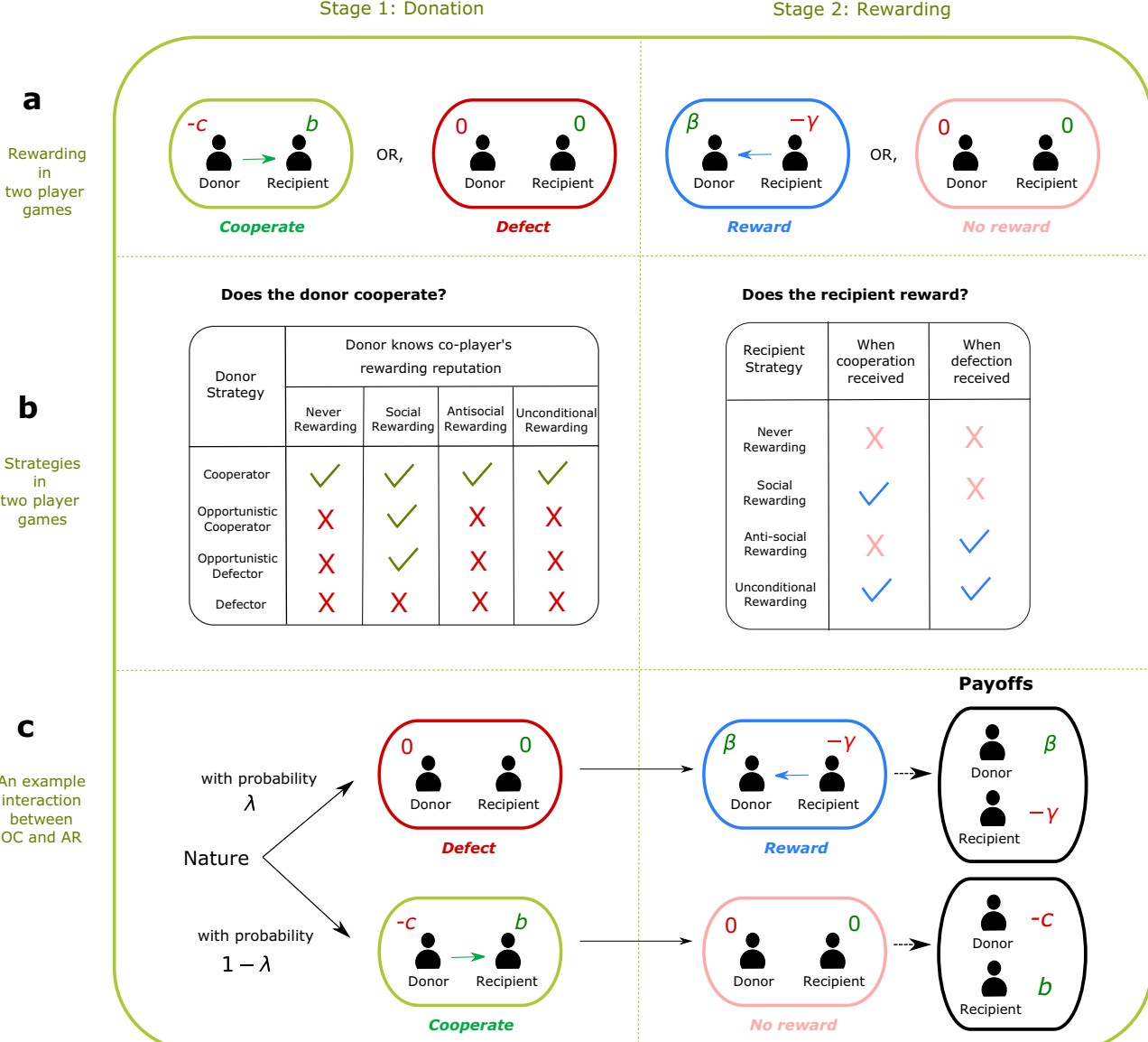

**Fig. 1 | Setup and strategies of the two-player donation game with rewarding.**
**a** For the baseline model, we consider interactions between two individuals, a donor and a recipient. First, the donor decides whether or not to pay a cost $c$ to provide a benefit $b$ to the recipient. We refer to the two possible actions as cooperation and defection, respectively. Afterwards, the recipient decides whether or not to reward the donor. Rewards come at a cost $\gamma$ to the recipient and yield a benefit $\beta$ to the donor. **b** Recipients can choose among four possible strategies (right panel). They either reward no one (NR for never rewarding), reward cooperators only (SR for social rewarding), reward defectors only (AR for antisocial rewarding), or reward everyone (UR for unconditional rewarding). To describe the donor's possible strategies (left panel), we assume that donors know a recipient's rewarding strategy with

probability $\lambda$. In that case, they can act opportunistically, by cooperating only with those recipients who reward socially. Overall, we distinguish four strategies for donors. They can either be unconditional cooperators (C), opportunistic cooperators (OC), opportunistic defectors (OD), and unconditional defectors (D). The two opportunistic strategies only differ in the way a donor acts when the recipient's strategy is unknown (in which case the donor may either cooperate or defect by default). **c** As an example, we illustrate a game between an OC-donor and an AR-recipient. With probability $\lambda$, the donor knows the recipient's strategy and hence defects (in which case the recipient rewards the donor). With probability $1-\lambda$, the donor does not know the recipient's strategy, and hence cooperates by default (in which case the recipient does not reward).

Overall, these results suggest that reputation effects should systematically favor both, cooperation and social rewarding. Importantly, there is no equilibrium in which recipients engage in anti-social rewarding. In fact, anti-social rewarding is weakly dominated by non-rewarding. Rather than rewarding defectors, it is better not to reward at all.

## Evolutionary analysis

Equilibrium analyses like the one above typically consider rational players. In a strict sense, these players are fully aware of all possible

payoffs, they perfectly understand which strategies are dominated, and they can assume their co-player to make similar inferences. For social interactions, however, it is perhaps more appropriate to assume that some of our behaviors are not consciously chosen, but rather the result of an evolutionary learning process. The results of such a learning process are often in close agreement with classical equilibrium predictions[55], as we also illustrate below.

We consider a well-mixed population of size $Z$. Population members are randomly matched to interact in the pairwise game described above. In any such interaction, the role of a given individual is randomly

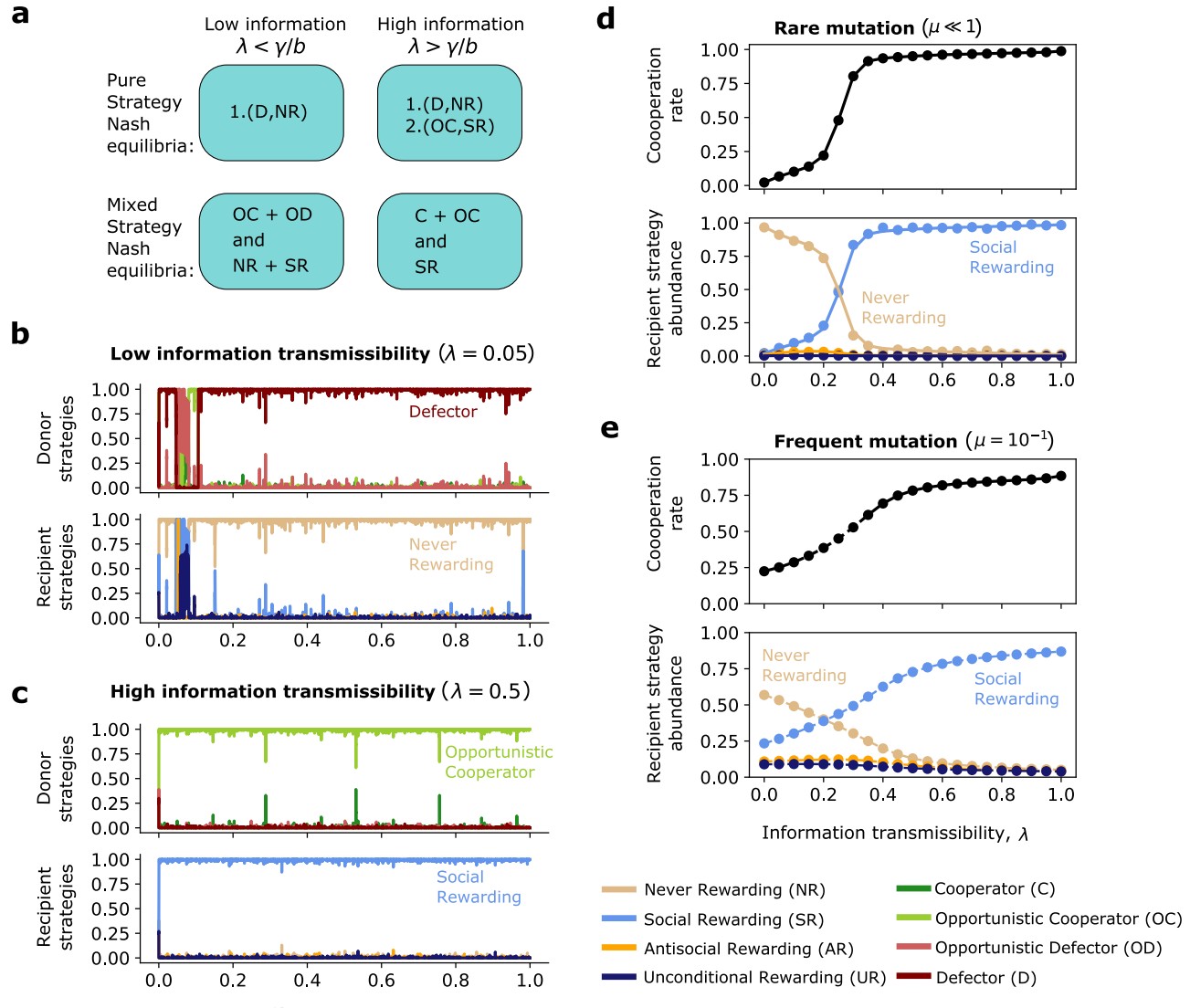

**Fig. 2 | Reputation effects facilitate the co-evolution of social rewarding and cooperation. a** To analyze the possible outcomes of the two-player game, we first describe all its Nash equilibria. This analysis shows that full cooperation can be sustained if $\lambda > \gamma/b$, such that socially rewarding recipients have sufficient opportunities to build up a reputation. **b**, **c** We expand on these static predictions by implementing evolutionary simulations for low and high information transmissibility. A low information transmissibility results in a population of non-rewarding defectors. A high information transmissibility results in a population in which individuals cooperate opportunistically, and reward socially. **d**, **e** These qualitative results neither depend on the exact information transmissibility $\lambda$, nor on the mutation rate $\mu$. Unless varied explicitly, we use the following default parameters for the simulations: Population size $Z = 100$, strength of selection $s = 1$, mutation rate $\mu = 10^{-4}$, and payoff parameters $b = 4$, $\beta = 3$, $c = \gamma = 1$. The dots in panels **d**, **e** show time averages over a simulation with $10^9$ time steps; the solid line in panel **d** represents the exact numerical solution in the limit of rare mutations[59].

determined; each individual is equally likely to act as a donor or as a recipient. As a result, the individuals' strategies take the form of a tuple $(\sigma_1, \sigma_2)$. Here, $\sigma_1$ is an individual's strategy when acting as a donor, and $\sigma_2$ is the strategy as a recipient. It follows that there are 16 possible strategies in total. Individuals are not restricted to keep their present strategies. Rather, they update their strategies through imitation and random exploration. To describe these updates formally, we use a pairwise comparison process[56]. At every time step, a randomly selected individual can change their strategy. With probability $\mu$ (the mutation rate), they adopt a random strategy different from their current one. With the converse probability $1-\mu$, they choose to update their strategy based on imitation. In that case, the focal player randomly samples a role model from the population. The probability that the focal player imitates the role model depends on the players' relative payoffs, and on a selection strength parameter $s$. Role models with a high payoff are more likely to be imitated (see the "Methods" section for details).

The results of this evolutionary process match the above equilibrium predictions. For small information transmissibilities, $\lambda < \gamma/b$, individuals learn not to reward anyone, and no one cooperates (Fig. 2b). Once information transmissibility exceeds this threshold, $\lambda > \gamma/b$, individuals quickly learn to reward socially, and to cooperate in response (Fig. 2c). These overall patterns are independent of the exact information transmissibility and of the considered mutation rate (Fig. 2d). In all cases, we observe that when recipients can gain a reputation, evolutionary processes select the equilibrium in which donors cooperate and recipients reward socially.

**Mutually beneficial rewards**
So far, we assumed that rewards are costly, $\gamma > 0$. In some instances, however, recipients may themselves benefit from rewarding their interaction partner. For example, the reward may consist in engaging in an activity that both parties benefit from. Such mutually beneficial

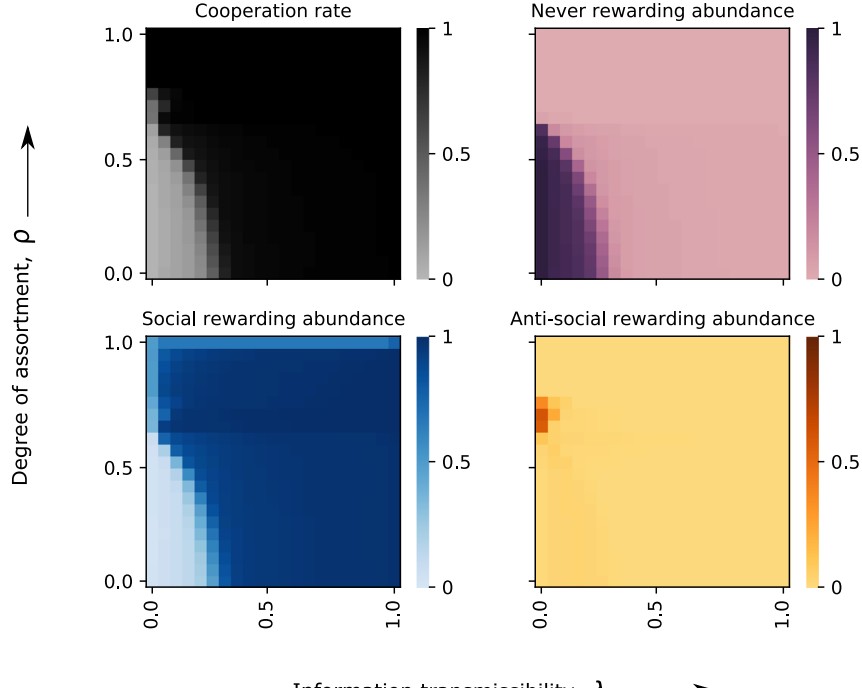

**Fig. 3 | Co-evolution of cooperation and rewarding in assorted populations.** We explore to which extent population structure affects the strategies that evolve in pairwise interactions. To this end, we systematically vary the population's information transmissibility $\lambda$ and the assortment parameter $\rho$. The limiting case of no assortment ($\rho = 0$) corresponds to the previously considered case of well-mixed populations. In the other limiting case of full assortment ($\rho \to 1$), individuals tend to only interact with co-players who use the same strategy. We find that both, high information transmissibility and high assortment, favor the evolution of cooperation and social rewarding. However, for $\lambda \approx 0$ and $\rho \approx 2/3$, we also identify a small parameter region in which anti-social rewarding can evolve. Here, defectors reward each other. The plots show the numerically exact solution in the limit of rare mutations[59]. Parameters are the same as in Fig. 2.

forms of rewards can be modeled by assuming that $\gamma < 0$. Here, recipients have strong incentives to reward the donor *in any case*, even if the donor defected. In fact, not to reward the donor would represent an instance of costly punishment: the recipient pays an opportunity cost of $-\gamma > 0$ to withhold a benefit $\beta > 0$ from the recipient. In the following, we explore whether recipients are still able to use rewards as a means to enforce cooperation when both parties actually prefer the recipient to reward.

First, we again characterize all (pure) Nash equilibria for $\gamma < 0$. We find two main classes of equilibria (see Supplementary Note 3 for proofs). The elements of the first class, (C, SR), (OC, SR), (D, AR), (D, UR), are always equilibria. The elements of the second class, (OD, AR) and (OD, UR), are equilibria when information transmissibility is low, $\lambda \leq -\gamma/(b-\gamma)$. In particular, while cooperation and social rewarding can still occur in equilibrium, we now find that also defection and antisocial rewarding are feasible. Further evolutionary simulations, however, suggest that instances of antisocial rewarding (and unconditional rewarding) are rare. They only emerge when the information transmissibility $\lambda$ is sufficiently small. If instead recipients have sufficient opportunities to build up a reputation, almost all of them learn to cooperate and to reward socially, despite their incentive to reward either behavior (Fig. S1).

## Co-evolution of cooperation and rewarding in assorted populations

Our previous analysis considers well-mixed populations. Every individual is equally likely to interact with every other. In most natural populations, however, there is some form of assortment[57]. As a result, individuals are more likely to interact with their own kind. Previous analyses of institutionalized rewards (without reputation effects) suggest that assortment can favor the evolution of antisocial rewarding[25–27], even when rewards are costly, $\gamma > 0$. In that case, populations converge to a state in which everyone defects, and all

population members reward each other for their selfish behavior. In the following, we explore to which extent such counterintuitive effects of assortment are ruled out when there is peer rewarding and recipients have a chance to build up a reputation.

To this end, we extend our model by introducing an assortment parameter $\rho \in [0, 1)$ (see the "Methods" section for an exact definition). For $\rho = 0$, we recover the previous case of a well-mixed population. As $\rho$ increases, individuals with strategy $(\sigma_1, \sigma_2)$ are increasingly likely to interact with population members who use the very same strategy. In the limiting case of $\rho \to 1$, individuals are guaranteed to interact with their own kind (provided the population contains at least one other member with that strategy). Such assortment can promote the evolution of dominated strategies[58]. As an example, consider a population in which $i$ individuals use (OD, NR) and $Z-i$ individuals use (OD, AR). While (OD, NR) players always get the higher payoff in well-mixed populations, (OD, AR) may yield the higher payoff in assorted populations. For this case to arise, the degree of assortment $\rho$ needs to be sufficiently large, and the number of non-rewarding players needs to be intermediate $i_1 < i < i_2$ (see Fig. S2). The exact thresholds $i_1$ and $i_2$ depend on $\rho$. They tend towards 0 and $Z$, respectively, as $\rho$ increases towards one.

To explore the impact of assortment on evolution, we consider simulations in which we systematically vary the assortment parameter $\rho$ and information transmissibility $\lambda$ between zero and one (Fig. 3). We find a small parameter region in which antisocial rewarding can prevail. In the absence of reputation effects, $\lambda \approx 0$, and for intermediate assortment, $\rho \approx 2/3$, a majority of individuals learn to reward anti-socially. This region slightly increases when rewards outweigh the benefit of cooperation, $\beta > b$ (Fig. S3). In general, however, assortment has a strongly positive effect on cooperation and social rewarding (Fig. 3). In fact, full cooperation can evolve even if there is no fully cooperative Nash equilibrium, provided that the assortment is sufficiently strong. Overall, we recover our previous finding that reputation

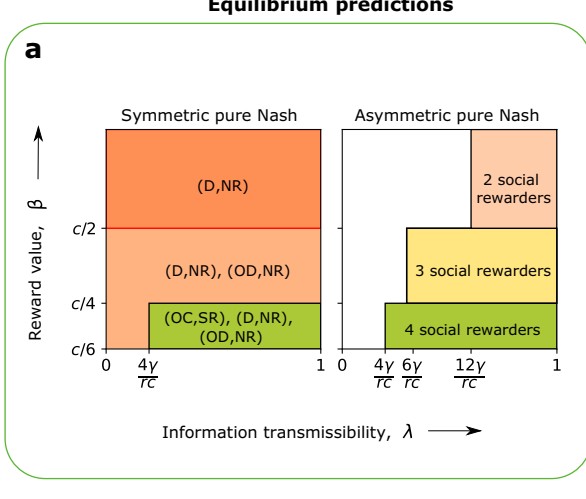

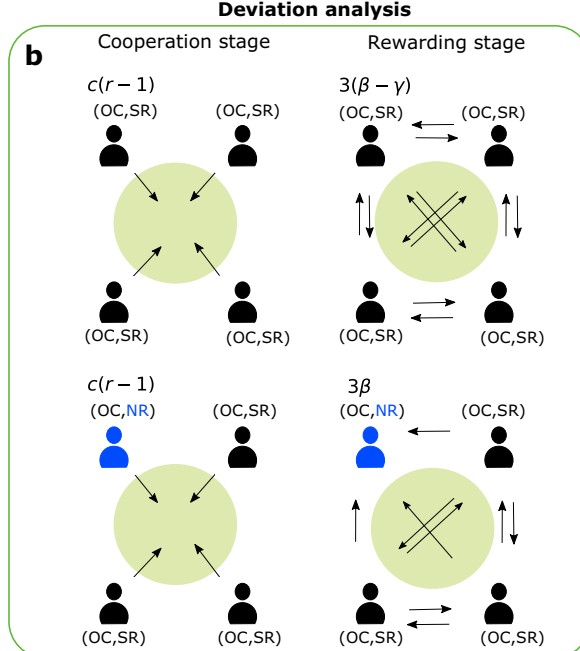

**Fig. 4 | Cooperation and rewarding in multiplayer interactions.** To explore how the previous results on pairwise games extend to larger groups, we consider public goods games with a subsequent rewarding stage. For the illustration, we consider groups of size $N = 4$. **a** Again, we first analyze the game's Nash equilibria. We describe both, all pure and symmetric equilibria (left panel), and all equilibria in which social rewarders and non-rewarders may co-exist (right panel, showing how many of the four group members are social rewarders in the respective equilibrium). In general, a high information transmissibility $\lambda$ makes it more likely that cooperation can be sustained in equilibrium. However, as rewards become increasingly effective (higher $\beta$), individuals may engage in second-order free riding. Such individuals contribute to the public good, but they do not reward others. **b** Here, we illustrate how second-order free riding can emerge. If $\beta$ is sufficiently large, a socially rewarding group member (top row) can deviate towards non-rewarding (bottom row). Other group members still find it worthwhile to cooperate, but the deviating group member saves the rewarding costs. **c** We explore the co-evolution of cooperation and rewarding by implementing additional evolutionary simulations. When mutations are rare (left panel), we find that rewarding may not need to evolve, even for large information transmissibilities $\lambda$. Here, second-order free riders completely destabilize (OC, SR) populations. However, when mutations are sufficiently frequent, (OC, SR) and (OC, NR) can stably coexist. A representative evolutionary trajectory that illustrates this latter case is illustrated in panel (**d**). The figure uses the same evolutionary parameters as Fig. 2 but with a public good multiplication factor $r = 2$ and a rewarding cost of $\gamma = 0.1$. In panel **d**, we additionally assume $\mu = 0.01$, $\lambda = 0.5$, and $\beta = 0.4$.

effects systematically favor social rewarding. Although anti-social rewarding can generally occur, it only evolves under rather restrictive assumptions. It requires nearly anonymous interactions (small $\lambda$) and intermediate assortment (an intermediate $\rho$).

## A model for the evolution of cooperation and rewards in multiplayer interactions

The previous results on pairwise games yield interesting insights into the interaction of cooperation and rewarding. Yet they do now allow us to study second-order free-riding: Only in larger groups, individuals may abstain from rewarding cooperation, hoping that someone else is

willing to reward cooperators. To explore how our previous results extend to larger groups, in the following we consider public goods games among $N$ players. Again, the game has two stages. The first stage is the contribution stage. Here, players decide whether or not to pay a cost $c > 0$ to make a contribution towards a common pool. Total contributions are multiplied by some productivity factor $r$, with $1 < r < N$. The resulting amount is equally shared among all group members. Similar to before, we refer to the act of contributing as cooperation and to not contributing as defection. The second stage is the rewarding stage. Here, after learning each other's contributions, players choose which of the other group members (if any) to reward.

For every rewarded group member, individuals pay a cost $\gamma > 0$ to provide a reward $\beta > 0$.

In analogy to the two-player game, the possible actions in the rewarding stage are to reward no one (NR), reward everyone who contributed (SR), reward everyone who did not contribute (AR), or reward all group members (UR). The players' rewarding behaviors again may become publicly known. Specifically, we assume that before entering the first stage, all individuals learn the correct rewarding strategy of all other group members with probability $\lambda$. With the converse probability $1-\lambda$, they do not learn anyone's rewarding strategy (more fine-grained models are possible; but to keep the analysis simple, we do not consider them here). Given the rewarding strategies in the second stage, one can easily derive when it is worth to contribute in the first stage: If an individual interacts with $m_{SR}$ social rewarders and $m_{AR}$ antisocial rewarders, cooperation in the first stage is profitable if and only if

$$\beta(m_{SR} - m_{AR}) \geq c\left(1 - \frac{r}{N}\right). \qquad (1)$$

That is, the effective gain from rewards (on the left-hand side) needs to offset the effective cost of cooperation (on the right-hand side). In particular, cooperation can only be profitable when there are more social rewarders than there are antisocial rewarders. Based on these considerations, we again consider four strategies for the first stage. Two strategies are unconditional, always cooperate (C) and always defect (D). The other two strategies act opportunistically. In case opportunistic players know the rewarding strategies of the other group members, they cooperate if and only if Eq. (1) holds. Otherwise, if the others' rewarding strategies are unknown, opportunistic players either cooperate (OC) or defect (OD). For this multiplayer game, deriving a general payoff formula is somewhat laborious. However, it is straightforward to compute payoffs algorithmically. We provide the respective code in our online repository.

Again, one can show that cooperation and any form of rewarding require some basic conditions to be feasible. On the one hand, rewards need to be sufficiently substantial to potentially warrant contributing to the public good, $(N-1)\beta \geq (1 - \frac{r}{N})c$. On the other hand, the benefit from the other group members' contributions needs to exceed the rewarding costs, $\frac{r}{N}c \geq \gamma$. If either of these two conditions is violated, there is no equilibrium in which group members cooperate, or in which they reward others (for proof and all further details, see Supplementary Notes 2 and 3). In the following, we thus assume that these two conditions are satisfied.

To analyze multiplayer interactions, we first characterize all pure and symmetric Nash equilibria (i.e., those equilibria in which every group member adopts the same deterministic strategy). Similar to the two-player case, we obtain two qualitative cases (Fig. 4a). The first case corresponds to defecting groups in which no one is rewarded. One instance of this case arises when individuals adopt (D, NR), which is always an equilibrium. The second case corresponds to a cooperating group in which everyone rewards socially, (OC, SR). Similar to the pairwise game, this case requires that individuals are sufficiently likely to learn each other's rewarding strategy, $\lambda \geq N\gamma/(rc)$. As a second condition, however, this equilibrium now also requires that rewards are not too profitable, $\beta < (1-r/N)c/(N-2)$. This latter condition prevents group members from becoming second-order free riders. Once rewards are too profitable, opportunistic group members find it worth to contribute even if not all other group members engage in social rewarding. As a result, a second-order free-riding problem arises: it takes some social rewarding to ensure mutual cooperation, but individuals prefer others to pay the respective rewarding costs. Once this second condition is no longer satisfied, players are incentivized to deviate towards (OC, NR), as illustrated in Fig. 4b. Stable cooperation is still feasible for sufficiently high values of $\lambda$, but now it requires an asymmetric equilibrium in which some individuals adopt (OC, SR) and others use (OC, NR). Interestingly, in any such mixed group, non-rewarders get a higher payoff than social-rewarders. Yet one can still show that neither type has an incentive to deviate if the information transmissibility is sufficiently large (see right panel in Fig. 4a for an illustration). Similar to the two-player case, anti-social rewarding is never part of any equilibrium (see Supplementary Note 2 for details).

We complement this static analysis by exploring the evolutionary dynamics of the public good game. We assume that individuals in a population of size $Z$ are randomly sampled to interact in groups of size $N$. As before, players adopt new strategies over time, either by mutation or by imitating other population members (see the "Methods" section). We find that the results depend on how abundant mutations are. When mutations are exceedingly rare[59–61], populations are homogeneous most of the time. Here, our simulations largely recover the outcomes predicted by the pure and symmetric Nash equilibria (Fig. 4c and Fig. S4). With respect to the reward $\beta$, there is only a small window in which full cooperation emerges. If rewards are too small, cooperation is not profitable. If they are too large, (OC, SR) populations are destabilized by second-order free riders (OC, NR). This outcome changes when mutations occur at an appreciable rate. Here, the learning dynamics quickly lead to mixed populations, in which (OC, SR) and (OC, NR) stably coexist (Fig. 4d). In this state of co-existence, both strategies obtain the same payoff in expectation. Non-rewarders have a payoff advantage in cooperative groups, as they do not pay any rewarding costs. On the other hand, rewarders have an advantage because they are more likely to end up in groups in which players cooperate in the first place (i.e., they are more likely to find themselves in a group in which condition (1) is satisfied for the other group members). In the co-existence equilibrium displayed in Fig. 4d, the two effects are in balance.

Overall, the multiplayer game thus leads to similar qualitative conclusions as the previous two-player model. Again, reputation effects have a crucial impact on whether or not cooperation evolves, and whether people reward each other. If reputation effects are sufficiently prominent, individuals learn to cooperate and to reward those who cooperate.

## Discussion

In this study, we revisit the literature on the interplay between cooperation and (peer) rewarding[17–28]. This literature explores whether individuals become more cooperative when others can compensate them for their cooperative behaviors. While some of the models we use build on previous work[21–24], the questions we ask are more elementary. Most previous models ask whether rewards can help groups to maintain cooperation. To tackle this question, the respective studies presume that all rewarding is social, and only cooperative individuals would ever be rewarded. In contrast, we ask what kinds of behaviors individuals may find worth rewarding in the first place. To this end, our model permits various rewarding strategies. Individuals may reward no one, only cooperators (social rewarding), only defectors (antisocial rewarding), or everyone. Which rewarding strategies evolve depend on the environment in which social interactions take place. If rewards are costly, populations are well-mixed, and individuals cannot build up a reputation for how they reward others, then neither cooperation nor any form of rewarding evolves. But if people may learn each other's rewarding strategies, or if there is assortment, cooperation and social rewarding evolve naturally (Figs. 2–4).

Antisocial rewarding is disfavored in most cases. However, there are two noteworthy exceptions. The first exception arises when rewards have positive payoff consequences for both parties. In this case of mutually beneficial rewards, individuals may be tempted to reward all group members, even those who act selfishly. As a result, populations may evolve into a state in which everyone defects, but defectors are rewarded anyway (Fig. S1). The second exception occurs

when there is assortment. Also here, we find parameter regions in which individuals defect and reward each other for defecting (Fig. 3 and Fig. S3). Under assortment, such an outcome is stable because defectors who stop rewarding antisocially become more likely to interact with population members who do not reward them either. A similar logic is at play when previous models describe the evolution of anti-social rewarding institutions in structured populations[25–27]. Both of these exceptions, however, only arise in the absence of reputation effects. Once individuals have opportunities to learn each other's rewarding strategies in advance (which seems particularly likely if rewarding is institutionalized), cooperation and social rewarding are favored (Fig. 3 and Fig. S1). Our results thus suggest that antisocial rewarding should be rare in most practical scenarios.

Similar reputation effects have also been shown to reduce anti-social punishment[51,52]. Both our model and these models on punishment have in common that reputation effects render higher-order incentives[40] unnecessary to sustain cooperation. In particular, in our model individuals cooperate because this increases the chance of getting a reward; conversely, individuals are willing to reward socially because this increases the chance that future interaction partners will cooperate. The importance of reputation has also been stressed by the literature on indirect reciprocity[62]. Interestingly, however, this literature focuses on a different kind of reputation. In indirect reciprocity, individuals gain a reputation for how they cooperate. In our model, individuals gain a reputation for how they react to other people's cooperation. The economic interaction that describes the rewarding stage does not need to resemble the interaction that describes the cooperation stage. In particular, while cooperation is by definition costly for the individual who cooperates, our model allows rewards to be beneficial for both parties.

It is sometimes suggested that criminal organizations, such as the mafia, represent an example of antisocial rewarding[25]. We hold a different view. Individuals in these organizations rarely reward each other for undermining social welfare per se. Rather they reward each other for taking actions that are beneficial for their own community (even if these actions are detrimental for the rest of society). According to Henrich and Muthukrishna, 'corruption, cronyism, or nepotism is really just cooperation on a smaller scale, often among relatives, friends, and reciprocal partners, at the expense of cooperation on a larger, impersonal scale'[63]. Therefore, we would argue that criminal organizations do not engage in antisocial rewarding; rather, they engage in a peculiar form of social rewarding. This example highlights a more general observation. Social rewarding does not necessarily promote behaviors that are beneficial for a population. It merely promotes behaviors that are beneficial for those who engage in social rewarding.

Similar to previous observations for punishment institutions[64], our results suggest that rewards are most effective when they act as a public signal. In that case, social rewarding can persuade future interaction partners that cooperation is in their own best interest. This observation may explain why rewarding opportunities enhance cooperation in some behavioral experiments but not in others. For example, Sefton et al.[65] reports for a public good game setting that rewards only increases cooperation temporarily. Similarly, Vyrastekova and van Soest[66] show that rewards are only effective when they enhance efficiency (i.e., $\beta/\gamma > 1$), but ineffective when they merely represent cash transfers ($\beta/\gamma = 1$). In both studies, rewarding decisions were made anonymously. Participants only learned whether or not they have been rewarded, but not the identity of the co-player who was willing to pay the respective cost. Such a design allows for a clear interpretation of rewarding as altruistic behavior[4]. Yet it reduces the participants' incentives to reward each other in the first place. In line with this interpretation, rewarding leads to more positive dynamics in experiments in repeated and non-anonymous settings[19,20].

While our model explores why individuals reward behaviors they benefit from, reputational benefits have also been reported when rewarding is administered by third parties[67], or when rewards are meant to compensate those individuals whose partner defected[68,69]. Future work could explore these seemingly more altruistic acts of rewarding. Similarly, our model uses a comparably coarse way to implement reputation effects. We assume that individuals learn each other's rewarding strategy with some fixed probability $\lambda$. Instead, future models could describe the reputation dynamics in more detail. Such models could incorporate, for example, that individuals are more likely to learn someone's rewarding policy the more often the respective individual has been observed rewarding others.

According to our study, social rewards need to be sufficiently profitable to sway others to cooperate. In terms of our model, this means that $\beta > c$ in pairwise interactions and $\beta > (1−r/N)c$ in the public good interaction. An interesting experiment suggests that participants understand the importance of these thresholds remarkably well when they need to figure out the minimum reward necessary to incentivize cooperation[70]. In the experiment, individuals engage in a series of trust games with changing interaction partners. The possible actions of player 1 correspond to the actions of donors in the pairwise game we studied (cooperate or defect). Similarly, the actions of player 2 can be mapped to a recipient's actions of not rewarding and social rewarding. Before deciding whether to cooperate, donors learn a sample of their recipient's past rewarding decisions. As in our game, donors only find it worthwhile to cooperate if their recipient is sufficiently likely to reward in response. The experiment shows that a substantial fraction of recipients reward sufficiently often for donors to cooperate, but not more often than necessary. Such recipients earn a larger payoff than the donors, although the game allows for equitable equilibria with equal payoffs. This finding resonates well with the general theme of our model. People do not necessarily use rewards to enhance fairness within their group. Instead, in some of their interactions, they may merely reward others to ultimately benefit themselves.

## Methods
### Payoffs of the two-player game
Based on the description of the two-player interaction in the main text, we can represent this game by two $4 \times 4$ payoff matrices $A$ and $B$. As in the main text, $\lambda$ is the probability that donors learn the rewarding strategy of the recipient they interact with. We use $\bar{\lambda} := 1 - \lambda$ to denote the converse probability of not knowing the recipient's rewarding strategy. Then the donor's possible payoffs are summarized in payoff matrix $A = (A_{ij})$,

$$
\begin{array}{c}
C \\ OC \\ OD \\ D
\end{array}
\begin{array}{cccc}
NR & SR & AR & UR \\
\left(\begin{array}{cccc}
-c & \beta - c & -c & \beta - c \\
-\bar{\lambda}c & \beta - c & \lambda\beta - \bar{\lambda}c & \beta - \bar{\lambda}c \\
0 & \lambda(\beta - c) & \beta & \beta \\
0 & 0 & \beta & \beta
\end{array}\right)
\end{array}
\tag{2}
$$

Similarly, the recipient's possible payoffs are summarized in the matrix $B = (B_{ij})$,

$$
\begin{array}{c}
C \\ OC \\ OD \\ D
\end{array}
\begin{array}{cccc}
NR & SR & AR & UR \\
\left(\begin{array}{cccc}
b & b - \gamma & b & b - \gamma \\
\bar{\lambda}b & b - \gamma & \bar{\lambda}b - \lambda\gamma & \bar{\lambda}b - \gamma \\
0 & \lambda(b - \gamma) & -\gamma & -\gamma \\
0 & 0 & -\gamma & -\gamma
\end{array}\right)
\end{array}
\tag{3}
$$

Based on these payoff matrices, we characterize all of the game's pure and mixed Nash equilibria[54], as illustrated in Fig. 2a. For the respective details, see Supplementary Note 1.

In addition to this static equilibrium analysis, we also explore the pairwise game with evolutionary simulations. To this end, we interpret it as a role game[55], played in a population of size $Z$. Members of the population may play the game in both roles, as a donor or as a

recipient, with equal productivity. Thus, individual strategies now take the form $\sigma = (\sigma_1, \sigma_2)$. Here, $\sigma_1 \in S_1 = \{C, OC, OD, D\}$ represents how the individual acts as a donor. Similarly, $\sigma_2 \in S_2 = \{NR, SR, AR, UR\}$ represents how the individual acts when in the role of the recipient. Suppose that at a given time point, the distribution of donor strategies in the population is given $(n_C, n_{OC}, n_{OD}, n_D)$, with $n_C + n_{OC} + n_{OD} + n_D = Z$. Similarly, suppose the distribution of recipient strategies is given by $(n_{NR}, n_{SR}, n_{AR}, n_{UR})$, with $n_{NR} + n_{SR} + n_{AR} + n_{UR} = Z$. Then the expected payoff of a player with strategy $(\sigma_1, \sigma_2)$ in a well-mixed population is given by

$$\bar{\pi}(\sigma_1, \sigma_2) = \frac{1}{(Z-1)} \left( \frac{1}{2} \sum_{j \in S_2} n_j \cdot A_{\sigma_1, j} + \frac{1}{2} \sum_{i \in S_1} n_i \cdot B_{i, \sigma_2} - \frac{1}{2} (A_{\sigma_1, \sigma_2} + B_{\sigma_1, \sigma_2}) \right)$$

(4)

The factor 1/2 indicates that individuals are equally likely to act as a donor or as a recipient in any given interaction. The last term in Eq. (4) takes into account that individuals cannot play the game with themselves. We provide a Python implementation of this payoff formula in our online repository.

In addition to well-mixed populations, we also study which strategies evolve when there is assortment. Instead of assuming that all players are equally likely to interact, here we assume here that matching probabilities depend on the players' strategies. To compute expected payoffs for assorted populations, consider a player with strategy $(\sigma_1, \sigma_2)$. When matching this player with a co-player, we assume that co-players with the same strategy are $\xi$ times more likely to be chosen, compared to co-players with a different strategy. Here, $\xi$ is a parameter that ranges from $\xi = 1$ (all co-players are equally likely to be chosen) to $\xi \to \infty$ (co-players with the same strategy are exceedingly more likely). Based on this parameter $\xi$, we can define the player's payoff as follows:

$$\bar{\pi}_\xi(\sigma_1, \sigma_2) = \frac{1}{(n_{\sigma_1, \sigma_2} - 1)\xi + \sum_{(i,j) \neq (\sigma_1, \sigma_2)} n_{i,j}}$$
$$\cdot \left( \frac{A_{\sigma_1, \sigma_2} + B_{\sigma_1, \sigma_2}}{2} (n_{\sigma_1, \sigma_2} - 1)\xi + \sum_{(i,j) \neq (\sigma_1, \sigma_2)} \frac{A_{\sigma_1, j} + B_{i, \sigma_2}}{2} n_{i,j} \right)$$

(5)

Here, $n_{i,j}$ is the number of individuals in the population that adopt strategy $(i, j)$ with $i \in S_1$ and $j \in S_2$. Again, this formula takes into account that players cannot interact with themselves. According to the formula, assortment only has an effect on the player's payoff if there is at least one other population member with the same strategy. For easier interpretation, we can map the parameter $\xi \in [1, \infty)$ to a parameter $\rho \in [0, 1)$, by using the transformation $\rho = 1 - 1/\xi$. After this transformation, $\rho = 0$ corresponds to well-mixed populations, whereas $\rho \to 1$ corresponds to maximally assorted populations.

## Payoffs of the multiplayer game

The computation of payoffs in the $N$-player public good game is conceptually similar to the pairwise game. First, we compute the payoff of a given player with strategy $(\sigma_1, \sigma_2) \in S_1 \times S_2$ for fixed group composition. To this end, suppose the group composition is described by a vector $\mathbf{m} = (m_{(i,j)})$. Here, each entry $m_{(i,j)}$ describes how many players with strategy $i \in S_1$ and $j \in S_2$ are among the other group members. In particular, $|\mathbf{m}| := \sum_{i,j} m_{ij} = N - 1$. For any given group composition $\mathbf{m}$, we can compute the payoff $\pi_\mathbf{m}(\sigma_1, \sigma_2)$ that the focal player would get in the respective group. Due to the complex strategy space, we consider, deriving an explicit expression for $\pi_\mathbf{m}(\sigma_1, \sigma_2)$ is somewhat laborious. However, payoffs are straightforward to compute algorithmically. We provide our code in the online repository.

Given the payoffs in fixed groups, we can also compute the players' expected payoffs when they interact in a well-mixed population[71,72]. Again, consider a population of size $Z$, and a focal player with strategy $(\sigma_1, \sigma_2) \in S_1 \times S_2$. Moreover, suppose the distribution of the remaining population is given by a vector $\mathbf{n} = (n_{(i,j)})$. Here, each entry $n_{(i,j)}$ gives the number of remaining population members that adopt strategy $(i, j) \in S_1 \times S_2$. In particular, $|\mathbf{n}| := \sum_{i,j} n_{ij} = Z - 1$. We consider the case that groups are formed randomly, by sampling $N - 1$ group members from the population without replacement. In that case, a player's expected payoff is given by the formula

$$\bar{\pi}_\mathbf{n}(\sigma_1, \sigma_2) = \frac{1}{\binom{Z-1}{N-1}} \sum_{|\mathbf{m}| = N-1} \prod_{(i,j)} \binom{n_{(i,j)}}{m_{(i,j)}} \pi_\mathbf{m}(\sigma_1, \sigma_2)$$

(6)

The above equation provides a convenient formula to compute payoffs in well-mixed populations.

## Evolutionary dynamics and simulations

To study the evolutionary dynamics, we use a pairwise comparison process[56]. The process takes place in a finite population of size $Z$. At any given time point, players are equipped with a strategy $(\sigma_1, \sigma_2) \in S_1 \times S_2$ to interact with the other population members. As a result, they derive a payoff that is either given by Eq. (4) (in the case of pairwise games in well-mixed populations), by Eq. (5) (pairwise games in assorted populations), or by Eq. (6) (multiplayer games in well-mixed populations). To incorporate learning, we randomly select a player from the population. With probability $\mu$ (the mutation rate), this player switches to a randomly selected strategy. With probability $1 - \mu$, the player randomly selects a role model from the population (all other population members have the same chance to be selected). Suppose the focal player's payoff is $\pi$ and the role model's payoff is $\pi'$. Then the probability that the focal player switches to the role model's strategy is given by a Fermi function[73,74], $p = (1 + \exp[-s(\pi' - \pi)])^{-1}$. The parameter $s \geq 0$ is the strength of selection. For $s \to 0$, selection is weak. Here, the switching probability is $p \approx 1/2$, irrespective of the payoffs of the two players. In the other limit $s \to \infty$, selection is strong. Here, there is only a positive switching probability if the role model yields at least the focal player's payoff.

The above process is straightforward to implement with simulations. We use the same basic process for both the baseline model and the model extensions (the extensions only differ in the way how payoffs are computed but not in how individuals adapt their strategies). For any given simulation run, we record how abundant different strategies are at each point in time, and how often players cooperate. To calculate the cooperation rate for a given population composition, we compute the average cooperation probability over all possible interactions in the respective population.

If we additionally assume that mutations are sufficiently rare, then numerically exact results are feasible[59–61]. In that case, the dynamics in the population can be represented as a Markov chain. The states of this Markov chain are all 16 homogeneous populations (in which every population member adopts the same strategy). Once a mutant arises, this mutant fixes or goes extinct before the next mutation occurs. When mutations are rare, the Markov chain approach is both more precise and computationally more efficient than the simulations. We thus use this approach, as described by Fudenberg and Imhof[59], when we compute results in the weak mutation limit (e.g., in Figs. 2d, 3, and 4c). In the weak mutation limit, cooperation rates are calculated by taking weighted averages over the cooperation rates in each homogenous population.

## Reporting summary

Further information on research design is available in the Nature Research Reporting Summary linked to this article.

## Data availability

The generated simulation data is available at zenodo[75] and on GitHub https://github.com/Saptarshi07/social-rewarding.

## Code availability

All simulations were performed with Python. The respective code is available at zenodo[75] and on GitHub: https://github.com/Saptarshi07/social-rewarding.

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

## Acknowledgements

We thank the members of the Max Planck Research Group 'Dynamics of Social Behavior' (DynoSoB) for helpful discussions and constructive feedback. We acknowledge generous funding from the European Research Council (ERC) under the European Union's Horizon 2020 research and innovation program (Starting Grant 850529: E-DIRECT), and from the Max Planck Society.

## Author contributions

S.P. and C.H. designed the research; S.P. performed the research; S.P. and C.H. wrote the paper.

## Funding

## Competing interests

The authors declare no competing interests.
