## [Peer Review File · Nature Communications]

Reputation effects drive the joint evolution of cooperation and social rewardingReviewers' Comments:

Reviewer #1:

Remarks to the Author:

This paper introduces a model addressing social rewarding and cooperation. Social rewarding is the “positive” cousin to punishment, i.e., individuals pay a cost to reward good behaviour. The authors do not ask if or when social rewarding favours cooperation, but rather ****when does rewarding evolve****, given that rewarding may carry repetitional effects. Therefore, the paper inspects the coevolution of rewarding and cooperation. The main result is that in most interesting cases, social rewarding is selected for together with cooperation. This is based on the idea that behaviours such as rewarding (and punishment) may be efficient because they are seen by others.

The model considers a game with two stages. In stage one, pairs of players play a donation game with an option to behave opportunistically if knowing the rewarding behaviour of the opponent — via some repetitional mechanism. This leads to four strategies: always cooperate, cooperate if ignoring the rewarding behaviour of the opponent but behave opportunistically otherwise, defect if ignoring the rewarding behaviour of the opponent but behave opportunistically otherwise, and always defect. Donors know the rewarding behaviour of the opponent with probability λ . In stage two, recipients apply their rewarding strategy, one of four possibilities including unconditional behaviour as well as social and antisocial rewarding.

The authors also offered a very complete study, with the following variations to their model: i) Mutually beneficial rewards, capture by a negative rewarding cost; ii) population structure given by an assortment parameter; and multiplayer interactions, beyond pairwise donations. The results for these variations are understandably nuanced. But cooperation and rewarding remain an equilibrium favour by dynamics across all variations. This means the results are very robust.

I also want to highlight the very sound methodology. The setup includes an unbiased and complete behavioural space, and therefore results are also unbiased — this should be standard practice in the field but it isn't. Moreover, the paper does a rigorous static equilibrium analysis before embarking on dynamics. This is important, because only in the presence of multiple equilibria does dynamics matter. This is rare, but frankly should also be standard practice when feasible. The computer program behind the calculation/simulation is also made public. The methodology here is very sound, but also exemplary.

I have only very few remarks. I wish the authors would discuss more the implications of this research for the literature on punishment, particularly when it may include higher order punishment. Does this model cover that case? What can we learn here about that open question? I would also encourage the authors to discuss what questions remain open given this work, and what is the relationship of this model to models of indirect reciprocity.

In conclusion, this is a great paper; the beautiful result that you only get when asking the right questions and using all the tools that are appropriate. I can envision many papers following from this.

Reviewer #2:

Remarks to the Author:

In "Reputation effects drive the joint evolution of cooperation and social rewarding" authors study a model to systematically study the co-evolution of cooperation and different rewarding policies. In the model, both social and antisocial behaviors can be rewarded, but individuals gain a reputation for the way they reward others. By characterizing the game's equilibria and by simulating evolutionary learning processes, research shows that reputation effects systematically favor the co-evolution of cooperation and social rewarding. The main model applies to pairwise interactions in well-mixed

populations, but similar conclusions are obtained also when there is assortment, or when individuals interact in larger groups.

Taken together, the results suggest that rewards are most effective when individuals apply them strategically, as a means to sway others to cooperate, and this is consistent with empirical observations showing that people reward others to ultimately benefit themselves.

Overall, I have very much enjoyed reading this paper. I find it comprehensive and clearly written, and introducing new, timely, and very interesting results that will surely also inspire future research along these lines, for example, as the authors note, for improving our understanding of why we reward others and also how other forms of prosocial behavior can be promoted with positive incentives.

I am in principal in favor of publication, but subject to the following revisions. The introduction falls short in giving credit to related preceding research where positive and negative incentives have been studied before in similar simple models. I would note Evolutionary establishment of moral and double moral standards through spatial interactions, Dirk Helbing, et al., PLoS Comput. Biol. 6, e1000758 (2010) for the consideration of antisocial strategies, and Synergistic third-party rewarding and punishment in the public goods game, Yin Hai Fang, et al., Proc. R. Soc. A 475, 20190349 (2019) for the consideration of different rewarding strategies. Finally, I note a review of spatially extended evolutionary games that have been studied with punishment, rewarding and both, namely Statistical physics of human cooperation, Matjaž Perc, et al., Phys. Rep. 687, 1-51 (2017). I can accept the authors' argument that some forms of rewarding they study have not been studied before, and that this merits the recognition that publication in Nature Communications confers, but previous efforts that have considered the same type of simple models, albeit with somewhat different strategy sets, should be properly acknowledged.

Also, it would be very useful if the authors would make their source code available as supplementary material. This would promote the usage of the proposed model and allow also others to take better advantage of this research, and also allow them to reproduce the results. A bit more details as to the simulation details, in particular for the extended games with assortment and group interactions would also be most welcome.

Apart from this, I am happy to congratulate the authors to an inspiring work.

Reviewer #3:

Remarks to the Author:

In the present paper, the author investigate a simple two-player, two-stage, gift-giving game to investigate under what condition reciprocal cooperation can develop. They specifically consider strategies that reward non-cooperation to see under what conditions "anti-social rewarding" can emerge and also extend the model to an N-person public goods setting.

From what I can tell, there are several interesting design choices, like considering anti-social reward strategies, assortment, a 'transmissibility' parameter that governs the knowledge that the first-mover has of the second-mover's response, and investigating the resulting conditional strategies (that the authors label opportunistic).

The paper is really well written, easy to follow, and addresses an interesting and important question. Namely, how social rewarding can foster cooperation and under what conditions anti-social rewarding can theoretically occur. The model is clearly presented, extended to the N-person case, and the dynamics are well described (with both simulations and equilibrium analyses).

As such, I have very little to criticize and all the points I list below are merely aimed at clarifying some concepts and their relation to previous literature and concepts. The model is interesting (especially the

generalization to the N-person case and the emergence of stable second-order free-riding) and, I could imagine, can inspire interesting future work both through extending the model or testing it empirically.

Hence, I would like to highly recommend it for publication.

Still, I had some clarification questions and suggestions that I list below.

1) In the abstract and the intro, the authors refer a lot to findings on (anti-social) punishment (e.g., "They reinforce cooperation by offering rewards, and they disincentivize defection by exerting various forms of punishment. Although such incentives facilitate the evolution of cooperation, the question remains how these incentivizing behaviors themselves evolve" in the abstract or "For example, there is by now a substantial literature suggesting that punishment options can be misused ...", page 1). Based on the abstract and intro, I was expecting a model that compares punishment and reward strategies. Yet, their model is only really considering reward strategies (except maybe with the extension described on page 5, bottom).

As such, the abstract and intro could be a bit misleading and the authors should clarify to which degree their model can really contribute to our understanding of misuse of punishment (or more clearly indicate that this paper is not aimed at understanding punishment at all).

2) The authors use parameter λ to denote the probability that the first mover knows the strategy of the second mover. At first, I was a bit puzzled by this concept. Why would the second-mover commit to an action? Isn't that the advantage of a second-mover that she can decide after the first-mover decided? It seems to blur the concept of first- and second-mover. With $\lambda = 1$, the roles are somewhat reverted – player 1 can condition her action based on player 2's strategy rather than the other way around.

From this perspective, it also becomes clear that player 1 would simply cooperate if (she knows that) player 2 rewards cooperation (and the benefits of reward outweigh the cost of cooperation) and player 2 always rewards if the benefits of cooperation outweigh the cost of rewarding.

From my reading, λ simply changes the structure of the game (in a way, player 2 has to pre-commit and loses the "second-mover" advantage) and this is what allows the evolution of reciprocal cooperation in the model.

The authors interpret λ as "population information transmission" or reputation. This is a very interesting idea. Yet, the authors could also highlight that, with this operationalization, the underlying structure of the game changes (i.e., it is strictly not a sequential game anymore with high λ in my understanding) and my suggestion would be to highlight this change in the structure of the game more as it more easily helps to understand why this parameter is so crucial for the emergence of an equilibrium of reciprocal cooperation.

3) Related to (2), the authors also refer to reputation. Yet, there is no image scoring mechanism and image scoring alone can lead to the evolution of cooperation (i.e., simply having conditional cooperation strategies based on the image score of the target). As such, if the authors consider λ as capturing something about reputation in the system, I was wondering why rewarding is needed in the first place to explain the evolution of cooperation (if simple image scoring can already do that, as we know).

4) I was a bit puzzled about the agent-based simulations. On page 4 bottom, the authors correctly state that they assume rationality in their equilibrium analyses. This is maybe my ignorance or lack of understanding, but why would the agent-based simulations lead to different conclusions? It was not clear to me why the learning process (under no assortment) should, theoretically, lead to different outcomes (or, from a different angle, why it not also implies rationality). Maybe the authors could clarify or briefly educate me in this regard.

5) related to (4), at several points in the manuscript, the authors conclude that rewarding and cooperation emerges out of self-interest (e.g., in the discussion "People do not necessarily use rewards to enhance the fairness within their group. Instead, they may merely reward others to

ultimately benefit themselves"). This is fine, of course, but it is also a bit obvious: In their model, cooperation and rewarding needs to benefit the agent otherwise it would not evolve. Hence, it must be true within their model. Yet, this does not mean that this must necessarily be the case outside of their model. As such, it would be worth highlighting that the authors consider a model in which cooperation can only develop out of self-interest which, however, does not 'proof' that social rewarding among humans necessarily is always motivated by self-interest.

6) One of the most interesting findings (at least for me), was the emergence of a stable co-existence of second-order free-riding and social rewarding (page 8 top). Yet, when reading the main text, it was not clear to me why rewarders can co-exist with non-rewarders (i.e., second-order free-riders) – the latter even getting a higher payoff (by definition).

The intuitive explanation on page 9 ("once rewards are too profitable, opportunistic group members find it worth to contribute even if not all other group members engage in social rewarding. As a result, a second-order free riding problem arises: individuals understand that it takes some social rewarding to ensure mutual cooperation, but they prefer others to pay the respective rewarding costs.") was very helpful in this regard. I would suggest to move something similar (maybe without the "individuals understand") to the main text.

Yet, still it would also be great to further explain why rewarders and non-rewarders can co-exist even though non-rewarding seems to dominate rewarding. How does that exactly work in the simulations? Are there some kind of cycles between cooperation and defection and rewarding and second-order free-riding that repeats across time?

First, we would like to thank the editor and the reviewers for their efforts. The three review reports have raised many stimulating observations, and they provided us with valuable feedback. In the meantime, we have addressed all the reviewers' comments. For further details, please see the point-by-point reply below.

When preparing our revised manuscript, we also noted that **Figure 4a** was incomplete. In the left panel on symmetric and pure Nash equilibria, two additional pure strategy equilibria were missing in the green parameter region. The existence of these equilibria should have become clear from the main text, but these equilibria should also be displayed in the figure. In the revised manuscript, we thus provide a correct version. All results and conclusions remain unaffected.

Reviewer #1:

This paper introduces a model addressing social rewarding and cooperation. Social rewarding is the "positive" cousin to punishment, i.e., individuals pay a cost to reward good behaviour. The authors do not ask if or when social rewarding favours cooperation, but rather ****when does rewarding evolve****, given that rewarding may carry repetitional effects. Therefore, the paper inspects the coevolution of rewarding and cooperation. The main result is that in most interesting cases, social rewarding is selected for together with cooperation. This is based on the idea that behaviours such as rewarding (and punishment) may be efficient because they are seen by others.

The model considers a game with two stages. In stage one, pairs of players play a donation game with an option to behave opportunistically if knowing the rewarding behaviour of the opponent — via some repetitional mechanism. This leads to four strategies: always cooperate, cooperate if ignoring the rewarding behaviour of the opponent but behave opportunistically otherwise, defect if ignoring the rewarding behaviour of the opponent but behave opportunistically otherwise, and always defect. Donors know the rewarding behaviour of the opponent with probability λ . In stage two, recipients apply their rewarding strategy, one of four possibilities including unconditional behaviour as well as social and antisocial rewarding.

The authors also offered a very complete study, with the following variations to their model: i) Mutually beneficial rewards, capture by a negative rewarding cost; ii) population structure given by an assortment parameter; and multiplayer interactions, beyond pairwise donations. The results for these variations are understandably nuanced. But cooperation and rewarding remain an equilibrium favour by dynamics across all variations. This means the results are very robust.

I also want to highlight the very sound methodology. The setup includes an unbiased and complete behavioural space, and therefore results are also unbiased — this should be standard practice in the field but it isn't. Moreover, the paper does a rigorous static equilibrium analysis before embarking on dynamics. This is important, because only in the presence of multiple equilibria does dynamics matter. This is rare, but frankly should also be standard practice when feasible. The computer program behind the calculation/simulation is also made public. The methodology here is very sound, but also exemplary.

Reply: We thank the reviewer for the positive assessment. We fully agree that unbiased strategy spaces should become the default in evolutionary game theory.

I have only very few remarks. I wish the authors would discuss more the implications of this research for the literature on punishment, particularly when it may include higher order punishment. Does this model cover that case? What can we learn here about that open question?

Reply: Thank you for bringing up this topic (it was also raised by reviewer #3). Our research is explicitly tailored to explore how cooperation and rewarding can co-evolve. We show that the two behaviors can reinforce each other when individuals gain a reputation for how they reward others. While we do not study the co-evolution of cooperation and punishment, we suspect that a similar mechanism may be at work there. In fact, previous work suggests that reputation effects can play a similar role for the simultaneous emergence of cooperation and (social) punishment behaviors (see, for example, Hilbe & Traulsen, Scientific Reports, 2012). This line of work also suggests that higher-order punishment (i.e., punishing non-punishers) is not necessary to maintain cooperation. Instead, in these models, individuals cooperate to avoid punishment; and conversely, individuals are willing to punish defectors to deter future interaction partners from defecting.

Changes: While we prefer to keep the focus of our manuscript on rewarding, we briefly mention the above observations on punishment in our revised discussion section.

I would also encourage the authors to discuss what questions remain open given this work, and what is the relationship of this model to models of indirect reciprocity.

Reply: These are great suggestions. While we believe our manuscript provides several important insights into what kinds of rewarding behaviors can evolve in natural populations, there are several questions our model does not tackle. Here are two examples.

- (i) In our model we assume that players know their co-player's strategy with some exogenous probability λ . Future work could model more explicitly how individuals form reputations with respect to their rewarding behaviors.
- (ii) Our model can explain why individuals find it worthwhile to reward a group member that has been cooperative towards them. In some instances, however, people even reward third parties, for social behaviors they did not personally benefit from (relatedly, people sometimes help a victim of anti-social behavior, even if they have not been directly involved in the respective interaction). It seems to us our model does not give a full account of these more "altruistic" forms of social rewarding.

With respect to the relationship between this model and models of indirect reciprocity, we agree that the two frameworks are certainly related. For example, both frameworks assume that individuals act socially because this positively affects their reputation (more precisely: it increases the chance that others will cooperate with them in future). At the same time, however, we believe there are crucial differences between the two approaches. In indirect reciprocity, people earn a reputation for how they cooperate. Which behaviors are regarded as good and bad might depend (in non-trivial ways) on the social norm employed in a population. In contrast, in our model people earn a reputation for how they react to other people's cooperation. Having a reputation of rewarding cooperators is beneficial independent of the social norm that the population uses to evaluate other social behaviors.

Changes: Thank you for raising these two issues (what are the open questions for the future, and what is the relationship between our model and indirect reciprocity). We now address both of these topics in our revised discussion section.

In conclusion, this is a great paper; the beautiful result that you only get when asking the right questions and using all the tools that are appropriate. I can envision many papers following from this.

Changes: Thank you very much for this encouraging feedback!

Reviewer #2:

In "Reputation effects drive the joint evolution of cooperation and social rewarding" authors study a model to systematically study the co-evolution of cooperation and different rewarding policies. In the model, both social and antisocial behaviors can be rewarded, but individuals gain a reputation for the way they reward others. By characterizing the game's equilibria and by simulating evolutionary learning processes, research shows that reputation effects systematically favor the co-evolution of cooperation and social rewarding. The main model applies to pairwise interactions in well-mixed populations, but similar conclusions are obtained also when there is assortment, or when individuals interact in larger groups.

Taken together, the results suggest that rewards are most effective when individuals apply them strategically, as a means to sway others to cooperate, and this is consistent with empirical observations showing that people reward others to ultimately benefit themselves.

Overall, I have very much enjoyed reading this paper. I find it comprehensive and clearly written, and introducing new, timely, and very interesting results that will surely also inspire future research along these lines, for example, as the authors note, for improving our understanding of why we reward others and also how other forms of prosocial behavior can be promoted with positive incentives.

Reply: Thank you, we appreciate the positive evaluation.

I am in principal in favor of publication, but subject to the following revisions. The introduction falls short in giving credit to related preceding research where positive and negative incentives have been studied before in similar simple models. I would note Evolutionary establishment of moral and double moral standards through spatial interactions, Dirk Helbing, et al., PLoS Comput. Biol. 6, e1000758 (2010) for the consideration of antisocial strategies, and Synergistic third-party rewarding and punishment in the public goods game, Yin Hai Fang, et al., Proc. R. Soc. A 475, 20190349 (2019) for the consideration of different rewarding strategies. Finally, I note a review of spatially extended evolutionary games that have been studied with punishment, rewarding and both, namely Statistical physics of human cooperation, Matjaž Perc, et al., Phys. Rep. 687, 1-51 (2017). I can accept the authors' argument that some forms of rewarding they study have not been studied before, and that this merits the recognition that publication in Nature Communications confers, but previous efforts that have considered the same type of simple models, albeit with somewhat different strategy sets, should be properly acknowledged.

Reply: Thank you for bringing these papers to our attention, we enjoyed reading them. We fully agree that we should give proper credit to all previous work on incentives and cooperation. Therefore, the three studies mentioned above are certainly relevant for the model we propose.

However, we would also like to point out some key differences between these studies and our work. The paper by Helbing et al explores the co-evolution of cooperation and punishment (it does not directly address rewards). Moreover, while the paper allows all individuals to punish others, only defectors can be subject to punishment (adapting the terminology of our model, this means that only social punishment is permitted, but not antisocial punishment).

The paper by Fang et al considers both, rewards and punishment. In their model, however, the individual population members cannot decide who to punish and who to reward. Rather the model considers some form of exogeneous punishment and rewarding mechanism. The way how this mechanism works is predetermined. Unlike in our model, this rewarding mechanism is not under selection pressure.

Finally, the review paper by Perc et al is a valuable resource that highlights several previous modeling approaches to reward and punishment. However, similar to the model by Helbing et al, the models

presented in that review article presuppose that all rewarding is social. In contrast, with our model we would like to explore which behaviors (social or antisocial) people would ever find worth rewarding.

Changes: Despite these differences, we fully agree that all three papers are related to our work, and hence they should be mentioned. We refer to these papers in our revised manuscript.

Also, it would be very useful if the authors would make their source code available as supplementary material. This would promote the usage of the proposed model and allow also others to take better advantage of this research, and also allow them to reproduce the results.

Reply: We fully support this concern for reproducible research. In fact, already for our previous version, we created an online repository that contains all our source code and the associated data (for the link, see our code and data availability statement at the end of the manuscript).

One reason why we prefer to share our code via an online repository rather than posting it in the Supplementary Information is to allow for updates in future (for example, to fix inconsequential bugs or to provide more detailed information on how to use the code). The histories of these updates (if any) would be naturally tractable. Interested readers would be able to see both the changes and the original versions of the code.

A bit more details as to the simulation details, in particular for the extended games with assortment and group interactions would also be most welcome.

Reply and Changes: Thank you for pointing out that the description of our simulations may not have been entirely clear. The basic setup of the evolutionary simulations for the extended games is in fact identical to the setup of our baseline model; just the payoffs are computed differently. In addition to clarifying this aspect, we also renamed the respective Methods section to “Evolutionary dynamics and simulations”, to draw more attention to the description of our simulations. Finally, we now also provide a more detailed description for how we calculate average cooperation rates at each time point (this may not have become fully clear in our original manuscript).

Apart from this, I am happy to congratulate the authors to an inspiring work.

Reply: Thank you very much!

Reviewer #3:

In the present paper, the author investigate a simple two-player, two-stage, gift-giving game to investigate under what condition reciprocal cooperation can develop. They specifically consider strategies that reward non-cooperation to see under what conditions “anti-social rewarding” can emerge and also extend the model to an N-person public goods setting.

From what I can tell, there are several interesting design choices, like considering anti-social reward strategies, assortment, a ‘transmissibility’ parameter that governs the knowledge that the first-mover has of the second-mover’s response, and investigating the resulting conditional strategies (that the authors label opportunistic).

The paper is really well written, easy to follow, and addresses an interesting and important question. Namely, how social rewarding can foster cooperation and under what conditions anti-social rewarding can theoretically occur. The model is clearly presented, extended to the N-person case, and the dynamics are well described (with both simulations and equilibrium analyses).

As such, I have very little to criticize and all the points I list below are merely aimed at clarifying some concepts and their relation to previous literature and concepts. The model is interesting (especially the generalization to the N-person case and the emergence of stable second-order free-riding) and, I could imagine, can inspire interesting future work both through extending the model or testing it empirically. Hence, I would like to highly recommend it for publication.

Reply: Thank you for this encouraging feedback. We particularly appreciate the detailed comments and suggestions below.

Still, I had some clarification questions and suggestions that I list below.

1) In the abstract and the intro, the authors refer a lot to findings on (anti-social) punishment (e.g., “They reinforce cooperation by offering rewards, and they disincentivize defection by exerting various forms of punishment. Although such incentives facilitate the evolution of cooperation, the question remains how these incentivizing behaviors themselves evolve” in the abstract or “For example, there is by now a substantial literature suggesting that punishment options can be misused ...”, page 1). Based on the abstract and intro, I was expecting a model that compares punishment and reward strategies. Yet, their model is only really considering reward strategies (except maybe with the extension described on page 5, bottom). As such, the abstract and intro could be a bit misleading and the authors should clarify to which degree their model can really contribute to our understanding of misuse of punishment (or more clearly indicate that this paper is not aimed at understanding punishment at all).

Reply and Changes: Thank you for making us aware of this possible ambiguity. After re-reading our manuscript, we agree with the reviewer that our abstract and intro could give the wrong impression that our article deals with both rewards and punishment. To avoid this ambiguity, we now omit the word “punishment” in our abstract entirely. However, because we believe it is important to provide at least some background on the literature on punishment, we have kept the respective paragraphs in the introduction. We have slightly rephrased these paragraphs to make sure there is no confusion about the core focus of our paper. Finally, (and also to respond to a related comment by reviewer #1), we briefly discuss the implications of our research for models on punishment in the revised discussion section.

2) The authors use parameter lambda to denote the probability that the first mover knows the strategy of the second mover. At first, I was a bit puzzled by this concept. Why would the second-mover commit

to an action? Isn't that the advantage of a second-mover that she can decide after the first-mover decided? It seems to blur the concept of first- and second-mover. With $\lambda = 1$, the roles are somewhat reverted – player 1 can condition her action based on player 2's strategy rather than the other way around. From this perspective, it also becomes clear that player 1 would simply cooperate if (she knows that) player 2 rewards cooperation (and the benefits of reward outweigh the cost of cooperation) and player 2 always rewards if the benefits of cooperation outweigh the cost of rewarding.

From my reading, λ simply changes the structure of the game (in a way, player 2 has to pre-commit and loses the “second-mover” advantage) and this is what allows the evolution of reciprocal cooperation in the model. The authors interpret lambda as “population information transmission” or reputation. This is a very interesting idea. Yet, the authors could also highlight that, with this operationalization, the underlying structure of the game changes (i.e., it is strictly not a sequential game anymore with high λ in my understanding) and my suggestion would be to highlight this change in the structure of the game more as it more easily helps to understand why this parameter is so crucial for the emergence of an equilibrium of reciprocal cooperation.

Reply: This is a great observation. We fully agree; the default version of the game (with $\lambda = 0$) can be interpreted as a sequential interaction in which the donor is the first mover and the recipient is the second mover. Standard backward induction suggests that this game only has an undesirable equilibrium. According to this equilibrium, the second mover never rewards (not even cooperators). As a result, the first mover has no incentive to cooperate.

The effect of positive λ is indeed to allow recipients to pre-commit to a rewarding policy. In the limiting case of $\lambda = 1$, the interaction becomes equivalent to a game in which the order of moves is reversed. Now, the recipient first chooses a rewarding policy, and then the donor can respond to this rewarding policy. Interestingly, both players benefit from this change in the game's structure.

Changes: We now provide an explanation along these lines in our revised manuscript. Thank you for providing us with this interpretation, we believe it is extremely useful.

3) Related to (2), the authors also refer to reputation. Yet, there is no image scoring mechanism and image scoring alone can lead to the evolution of cooperation (i.e., simply having conditional cooperation strategies based on the image score of the target). As such, if the authors consider λ as capturing something about reputation in the system, I was wondering why rewarding is needed in the first place to explain the evolution of cooperation (if simple image scoring can already do that, as we know).

Reply: In this manuscript we aim to explore what motivates individuals to use positive incentives (rewards). We show that when others may learn (and adapt to) an individual's rewarding policy, people have an incentive to specifically reward cooperators. Of course, the literature on the evolution of cooperation knows of many other mechanisms that promote cooperation, including the mechanism of indirect reciprocity (image scoring). In indirect reciprocity, individuals gain a reputation for whether or not they cooperate. In contrast, in our model, individuals gain a reputation for how they react to the cooperation of others. These two kinds of reputations are related. However, it is the second kind of reputation that is more relevant to explain what motivates people to (directly) reward others.

Let us also mention that while indirect reciprocity is an important mechanism for cooperation, the specific rule of image scoring has been under debate. As some scholars have pointed out (e.g., Leimar and Hammerstein, “Evolution of cooperation through indirect reciprocity”, Proc. R. Soc. B, 2001), image scoring is unstable. Intuitively, this instability arises because individuals have no incentive to withhold cooperation from defectors (doing so would harm their own reputation). This has led researchers to explore more sophisticated forms of indirect reciprocity. These norms distinguish between “justified defections” and “unjustified defections” (e.g., the work by Ohtsuki & Iwasa, “How should we define

goodness”, JTB 2004). The resulting norms provide a more nuanced view of indirect reciprocity. In particular, it is no longer clear whether or not cooperation should always result in a reputation advantage (for example, a donor who cooperates with a “bad” recipient may arguably deserve a “bad” reputation). In contrast, the reputational consequences of rewarding are more straightforward. In our model, individuals always benefit from being known to reward cooperators.

Changes: We now briefly address the relationship between our model and models of indirect reciprocity in our revised discussion section (also to take into account a similar comment by reviewer #1).

4) I was a bit puzzled about the agent-based simulations. On page 4 bottom, the authors correctly state that they assume rationality in their equilibrium analyses. This is may be my ignorance or lack of understanding, but why would the agent-based simulations lead to different conclusions? It was not clear to me why the learning process (under no assortment) should, theoretically, lead to different outcomes (or, from a different angle, why it not also implies rationality). Maybe the authors could clarify or briefly educate me in this regard.

Reply: We believe our agent-based simulations are useful for two reasons.

First, these simulations illustrate that our static equilibrium analysis can also be used to understand the dynamics in evolving populations. Of course, to experts in game theory this observation may be trivial. After all, there are several results that connect the fixed points of evolutionary dynamics to the game’s Nash equilibria (the most well-known result of this kind is concerned with the replicator equation, see for example Theorem 7.2.1 in the book by Hofbauer and Sigmund). Nevertheless, we have made the experience that scholars from fields like psychology or biology often view equilibrium results with suspicion. To them, game theory seems to make unrealistically strong assumptions on the individuals’ cognitive abilities (e.g., individuals are assumed to recognize and avoid dominated strategies, they are assumed to anticipate that their co-players will do the same, etc). For scholars from these fields, it may be reassuring to see that natural learning processes lead to similar outcomes. These learning processes do not require individuals to have a detailed understanding of the game, or of the other population members. Instead, individuals are merely required to optimize their payoff over time by imitation. From the outset, it is not obvious that such a dynamics lead to a Nash equilibrium (and in fact, classical examples like the rock-scissors-paper game show that convergence is not guaranteed).

Second, we believe that simulations can provide additional insights because they can be regarded as “equilibrium selection devices”. For many parameter values, our model allows for multiple Nash equilibria. For example, if the population’s information transmissibility λ is sufficiently large, both (D,NR) but also (OC,SR) are a Nash equilibrium. In such a situation, it is natural to ask: Which equilibrium is more salient? Evolutionary dynamics can be one way to approach this question, by asking: Which equilibrium is more likely to be discovered by evolving populations? In this sense, our simulations allow us to conclude that (OC,SR) is the more salient equilibrium in a large region of the parameter space.

Changes: We have revised our manuscript to include the above arguments.

5) related to (4), at several points in the manuscript, the authors conclude that rewarding and cooperation emerges out of self-interest (e.g., in the discussion “People do not necessarily use rewards to enhance the fairness within their group. Instead, they may merely reward others to ultimately benefit themselves”). This is fine, of course, but it is also a bit obvious: In their model, cooperation and rewarding needs to benefit the agent otherwise it would not evolve. Hence, it must be true within their model. Yet, this does not mean that this must necessarily be the case outside of their model. As such, it would be worth highlighting that the authors consider a model in which cooperation can only develop

out of self-interest which, however, does not ‘proof’ that social rewarding among humans necessarily is always motivated by self-interest.

Reply and changes: When we stressed that rewarding and cooperation emerge out of self-interest, what we had in mind is the following. Both rewards and even more so punishment are often interpreted as altruistic behaviors (in fact, one of the most influential articles in the field is “Altruistic punishment in humans”, Fehr & Gächter, Nature 2002). These behaviors are seen as altruistic because (i) these behaviors help communities to become more cooperative, and (ii) they are individually costly for the one who punishes or rewards. We believe a model like ours can rationalize these seemingly irrational behaviors; when people reward others in their everyday interactions, these rewards do not only have (immediate negative) payoff consequences, but also (positive) reputational consequences. Of course, such models do not show that social rewarding among humans is always motivated by self-interest. However, it resolves the puzzle why people may wish to reward others at all (even if they have no further interactions with the respective individual). Related to these observations, it may also be worth noting that our model does not require that individuals opt for social rewarding as a conscious (strategic) choice. Instead, our simulations may be interpreted as saying that social rewarding can emerge as the result of a (possibly subconscious) imitation process.

Having said that, we fully agree with the reviewer that the respective statements may almost appear tautological. To avoid any confusion, we have rephrased the statements accordingly.

6) One of the most interesting findings (at least for me), was the emergence of a stable co-existence of second-order free-riding and social rewarding (page 8 top). Yet, when reading the main text, it was not clear to me why rewarders can co-exist with non-rewarders (i.e., second-order free-riders) – the latter even getting a higher payoff (by definition). The intuitive explanation on page 9 (“once rewards are too profitable, opportunistic group members find it worth to contribute even if not all other group members engage in social rewarding. As a result, a second-order free riding problem arises: individuals understand that it takes some social rewarding to ensure mutual cooperation, but they prefer others to pay the respective rewarding costs.”) was very helpful in this regard. I would suggest to move something similar (maybe without the “individuals understand”) to the main text.

Reply and Changes: Thank you for this suggestion. We are of course happy to move this explanation from the Supplementary Information to the main text.

Yet, still it would also be great to further explain why rewarders and non-rewarders can co-exist even though non-rewarding seems to dominate rewarding. How does that exactly work in the simulations? Are there some kind of cycles between cooperation and defection and rewarding and second-order free-riding that repeats across time?

Reply: We agree, our finding that rewarders and non-rewarders may stably co-exist may appear counterintuitive. After all, in any cooperative group that contains a mixture of rewarders and non-rewarders, the non-rewards obtain the higher payoff.

To make sense of this co-existence, it is worth to keep in mind the exact setup of these evolutionary simulations. In the simulations, we consider a large population (typically the population size is $Z=100$). From this population, groups of smaller size are formed randomly (we usually consider the case $N=4$). Suppose now the population consists of a mixture of rewarders and non-rewarders, specifically of players with strategies (OC,SR) and (OC,NR).

In this setup, (OC,NR) players have an advantage whenever they find themselves in a mixed group that contains sufficiently many social rewarders (such that condition (1) in the main text is satisfied). In that case, they receive the benefits of cooperation without having to pay for the rewarding costs.

On the other hand, (OC,SR) players have an advantage because they are more likely to find themselves in such a cooperative group in the first place. That is, everything else kept equal, being a social rewarder makes it more likely that the other group members will cooperate, because it makes it more likely that condition (1) is satisfied for the other group members.

The relative size of these two advantages depends on the fraction of social rewarders in the population (i.e., on how likely condition (1) is satisfied when randomly sampling co-players for a given player). In particular, there is a critical fraction such that the two advantages are exactly balanced. This stable co-existence is what is displayed in **Figure 4d**. Such trajectories that lead to a stable co-existence are by far the most common outcome in this parameter region.

We note that in some of our simulations, we also observe (temporary) instances of cycling. One such instance is shown below. In this case, we observe a trajectory in which (D,NR) populations tend to be stochastically invaded by (OD,SR), which in turn can lead to the invasion of (OC,SR), which in turn can be (neutrally) invaded by (C,SR), which quickly leads back to (D,NR). However, while such cycles are possible, they seem to be extremely rare. In fact, we only observed one such trajectory, out of 150 independent simulations that we ran for the evolutionary process

Figure: Evolutionary cycles in a simulation of the public good game. Parameters: $s = 1$, $c = 1$, $r = 2$, $\beta = 0.4$, $\gamma = 0.1$, $\lambda = 0.5$, $\mu = 0.01$.

Changes: We have added a short paragraph to the main text in which we explain in more detail why rewarders and non-rewarders may co-exist. While we believe the cycles displayed above are interesting, we feel that they are not sufficiently representative to be particularly highlighted.

Reviewers' Comments:

Reviewer #1:

Remarks to the Author:

Thank you for the thorough response. I am satisfied with all comments and changes from the authors.

Reviewer #2:

Remarks to the Author:

The authors have revised their manuscript comprehensively and with love to detail. I warmly recommend publication in present form.

Reviewer #3:

Remarks to the Author:

The authors addressed all my previous concerns and comments. I very much appreciate their extensive response and the corresponding changes they made in the paper.

As such, I have no remaining comments and would like to congratulate the authors for this elegant work on the co-evolution of cooperation and reward that, I think, will also inspire interesting work in the future.